# Healthy and Unhealthy Food Consumption in Relation to Quality of Life among Finnish Female Municipal Employees: A Cross-Sectional Study

**DOI:** 10.3390/nu14173630

**Published:** 2022-09-02

**Authors:** Elina Bergman, Henna Vepsäläinen, Maijaliisa Erkkola, Marika Laaksonen, Hannu Kautiainen, Markus A. Penttinen, Päivi Rautava, Päivi E. Korhonen

**Affiliations:** 1Department of General Practice, Institute of Clinical Medicine, University of Turku and Turku University Hospital, 20014 Turku, Finland; 2Department of Food and Nutrition, University of Helsinki, 00014 Helsinki, Finland; 3Fazer Group, 00941 Helsinki, Finland; 4Unit of Primary Health Care, Kuopio University Hospital, 70210 Kuopio, Finland; 5Folkhälsan Research Centre, University of Helsinki, 00290 Helsinki, Finland; 6Suomen Terveystalo, 20520 Turku, Finland; 7Department of Public Health, University of Turku and Turku University Hospital, 20014 Turku, Finland; 8Clinical Research Centre, Turku University Hospital, 20521 Turku, Finland

**Keywords:** quality of life, healthy diet, healthy lifestyle, prevention, occupational health

## Abstract

Aspects of good quality of life (QoL) have been found to motivate people to make lifestyle changes. There is also evidence that certain dietary patterns are associated with QoL. The aim of this work was to examine whether consumption frequencies of healthy and unhealthy food items are associated with QoL in female employees. A cross-sectional study was conducted among 631 Finnish female employees (mean age 49 years, SD = 10) from 10 municipal work units in 2015. Information about the participants was collected by physical examination, laboratory tests, self-administered questionnaires, including the Food Frequency Questionnaire (FFQ), and from medical history. QoL was assessed with the EUROHIS-Quality of Life 8-item index. A significant positive association was seen between consumption frequency of healthy foods and the EUROHIS-QOL mean score (*p* = 0.002). The association was small but comprehensive, also involving most dimensions of QoL. The consumption frequency of unhealthy foods was not associated with QoL. These findings are relevant when designing diet counselling, since QoL is an outcome that has been found to motivate people to change their health habits. Recommending abundant use of healthy foods could be a simple and convenient way of diet counselling at many health care appointments, where time consuming approaches are difficult to conduct.

## 1. Introduction

The benefits of healthy eating, for example, according to the Mediterranean diet, have been indisputably shown in the prevention and treatment of several noncommunicable diseases [1,2,3]. Information about a healthy diet is available, and diet counselling is performed especially in primary health care settings where health promotion and prevention are the cornerstones of the work. However, nutrition recommendations may sometimes seem quite challenging, and complying with the advice is difficult for the patients [4]. It is known that the risk for non-adherence increases if the behavioural recommendations are complex [5], particularly since physicians’ appointments are often too short for comprehensive diet counselling. Simple and easily delivered ways of giving nutritional advice would therefore be needed.

There is evidence that positive changes in health behaviours, including changes in diet, can improve subsequent well-being [6]. In addition, QoL and well-being are outcomes that are found to motivate people to change their health habits [7,8]. It would be essential to know the aspects in nutrition that are the most importantly associated with QoL to be able to design very simple, yet effective diet counselling. We should know if it would be better to advise the patients to avoid certain unhealthy foods, or rather to recommend abundant use of some healthy foods, to obtain benefits in well-being.

In the previous literature, there is information about how certain dietary patterns are linked with QoL [9,10,11,12,13,14]. The most frequently studied is the Mediterranean diet, which has been associated with better self-reported physical and mental health in several studies [10,11,12,14]. In addition, other dietary patterns that are regarded as healthy have been found to be associated with better QoL [9,13]. Some studies have also found a negative association between unhealthy “Western” dietary patterns and QoL [11].

The dietary choices are not independent of each other, and an unhealthy dietary pattern typically contains abundantly unhealthy foods and simultaneously scarcely healthy foods [11,15]. It may not be possible to separate these two aspects completely, but by comparing different dietary patterns, no information is obtained about what aspects are responsible for the significant difference. There are, however, also studies about how some food groups are associated with QoL. Especially, frequent consumption of fruit and vegetables has been associated with better QoL in different populations [16,17,18]. Regular fish intake has been associated with better mental health measures [19], and high-fibre diet has been linked with better self-reported physical and mental health as well as increased energy [20]. In a few studies among children and young adults, there is some evidence that frequent consumption of unhealthy foods, like fast food and snacks, is associated with lower QoL [18,21]. Few studies have, however, studied the intake of foods, divided into healthy and unhealthy food groups, in association with QoL. To our knowledge, there is no information about how different consumption frequencies of healthy or unhealthy food groups are associated with QoL among working-age women.

The aim of this work was to examine the association of the daily consumption frequencies of healthy and unhealthy food groups with QoL in female municipal employees. We hypothesised that a frequent consumption of healthy food groups would have a positive relationship with QoL. If this hypothesis was proven correct, it could help in designing diet counselling for female employees.

## 2. Materials and Methods

### 2.1. Participants

This cross-sectional study was part of the PORTAAT (PORi To Aid Against Threats) study conducted among employees of the city of Pori (83,497 inhabitants in 2014) in south-western Finland in 2014 and 2015. The participating work units were selected by the chief of the Welfare Unit of Pori. Invitations to participate and information about the study were sent to employees via e-mail by the managers of the selected ten work units (total number of employees: 2570). Information events were also organised for the employees. The employees willing to participate contacted the study coordinator at their work unit, who then sent their contact information to the study nurse. There were no exclusion criteria. A total of 836 employees (104 males, 732 females) chose to participate in the study in 2014 (response rate 32.5%). The data collection was performed in October–December in 2014. First, the participants attended an enrolment visit with the study nurse, where an informed consent to participate was gathered from all the participants, and the study questionnaires were given to be completed at home before the next study visit. At the second visit, the study nurse collected the completed questionnaires and performed the height, weight, and blood pressure measurements.

The gender distribution of the initial respondents in 2014 (88% females) resembles the gender distribution among the employees of the work units that were invited to the study (86% females in 2014), and is close to the distribution among the employees of the city of Pori (78% females in 2014) [22]. It is also close to the gender distribution among Finnish public sector employees in general (around 80% females) [23]. The mean age among the initial study participants resembled the mean age among the whole personnel of the city of Pori [22], and the mean annual rate of sickness absence days did not vary significantly between the study participants and the non-participants on the included employment sectors [24,25].

All the initial respondents were invited to the second part of the study in October–December in 2015, and 710 (85%) of them attended (79 males, 631 females). The invitations to the follow-up visit, and the questionnaires to be completed at home before the visit, were sent to the participants by e-mail, or on request, by mail. At the visit, height, weight, and blood pressure were again measured by the study nurse, who also collected the completed questionnaires. The gender distribution at follow-up (89% females) was similar compared to baseline, and the proportions of different levels of vocational education were also unchanged, with 45% having university-level education. In the present work, all the information is from the year 2015 because data from the food frequency questionnaire (FFQ) was available only from that year. Since there were so few male participants, and it is also known that nutritional behaviours vary between genders [26], we restricted the analyses only to the women. For the present analyses, we report data of the 631 female participants who completed the follow-up study in the year 2015. The participants’ occupations included librarians, museum employees, janitors, IT workers, social workers, nurses, physicians, administrative officials, and general office staff.

### 2.2. Quality of Life

QoL was assessed with the EUROHIS-QOL 8-item index [27]. It is a shortened version of the WHOQOL-BREF scale, a widely used instrument for the assessment of generic QoL [28,29]. The domains in both questionnaires are the general, physical, psychological, social and environmental aspects of QoL. In EUROHIS-QOL there are two items about overall QoL and general health, and two items from the domains of physical health and environmental aspects. There is one item about psychological aspects as well as social relationships. The EUROHIS-QOL instrument has good internal consistency (Cronbach alpha: 0.80) and convergent and discriminant validity tested in several countries and different populations [30,31]. It has been shown that EUROHIS-QOL has retained the good psychometric properties of its parent, WHOQOL-BREF [31]. The participants of the present study answered the questions at home before the study visits. Every question was scored from one (very poor) to five (very good). All scores were then added together and divided by eight (the sum of the questions) to obtain the EUROHIS-QOL mean score [30].

### 2.3. Physical Examination

Height and weight were measured by a study nurse with subjects in standing position without shoes and outer garments. Weight was measured to the nearest 0.1 kg with calibrated scales and height to the nearest 0.5 cm with a wall-mounted stadiometer. Body mass index (BMI) was calculated as weight (kg) divided by the square of height (m^2^). Blood pressure (BP) was measured by a study nurse with an automatic validated blood pressure monitor with subjects in a sitting posture, after resting at least 5 min. Two readings taken at intervals of at least 2 min were measured, and the mean of these readings was used in the analysis.

### 2.4. Laboratory Tests

Laboratory tests were determined using blood samples which were obtained after at least 8 h of fasting. Plasma glucose, total cholesterol, high-density lipoprotein cholesterol (HDL-C), and triglycerides were measured enzymatically (Architect c4000/c8000). Low-density lipoprotein cholesterol (LDL-C) was calculated by the Friedewald’s formula [32]. All laboratory assays were performed in a single laboratory.

### 2.5. Psychological Symptoms

Depressive symptoms were assessed with the Major Depression Inventory (MDI) questionnaire [33], which can be used as an assessment tool for the severity of depressive symptoms. A total score of 0–20 is considered as no symptoms, 21–25 as mild symptoms, 26–30 as moderate symptoms, and 31–50 as severe depressive symptoms [34]. Anxiety was assessed with the General Anxiety Disorder 7-item Scale (GAD-7) [35]. In the GAD-7, a total score of 0–4 is considered as no anxiety, 5–9 as mild anxiety, 10–14 as moderate anxiety, and 15–21 as severe anxiety [35].

### 2.6. Health Behaviours and Other Measures

Physical activity (PA) was assessed with a self-administered questionnaire about the frequency and duration of leisure-time physical activity and commuting activities in a typical week. Both moderate aerobic PA (e.g., walking) and vigorous PA (e.g., running) were considered, and PA was reported as hours per week. Smoking status was assessed by a questionnaire. Alcohol consumption was assessed using the three-item Alcohol Use Disorders Identification Test (AUDIT-C) [36]. Sleep quality was assessed with the question, “During the past month, how would you rate your sleep quality overall?” (very good, good, poor, or very poor). In the analyses, the two highest classes of sleep quality were combined and set to indicate good sleep quality. Information about age, education years, marital status (cohabiting or not), and previously diagnosed chronic diseases was gathered using self-administered questionnaires and medical records.

### 2.7. Food Frequency Questionnaire

The participants reported their food consumption frequencies during the past week using a 45-item FFQ. The FFQ was based on the version developed for and validated to study children’s diets with parents filling in the questionnaires [37], and with specific attention paid to capture the consumption patterns of vegetables and fruit as well as sugary foods and beverages. A shortened, 25-item version of the FFQ has been tested for reproducibility with mostly moderate or good intraclass correlation coefficients [38]. The PORTAAT FFQ included eight food groups: vegetables, fruit, and berries; dairy products; fats and oils; fish; meat and eggs; cereal products; drinks; and others (i.e., sweets and snacks). Additional lines for the frequencies of meals and the use of dietary supplements were included. The FFQ had three answer columns: not at all, times per week and times per day. The instruction was to either tick the not at all column or to write a number in one of the other columns.

The items in the eight main categories of the FFQ were categorized into healthy and unhealthy foods, and eleven groups of healthy foods and six groups of unhealthy foods were derived (Table 1) based on the Nordic Nutrition Recommendations [39] for healthy and balanced diet and on previous scientific understanding about the association between diet and cognitive health [40,41]. Nutritionally neutral items were excluded. From the data collected with the FFQ, we summed the mean daily consumption frequency of foods in the same group and used that as an indicator of dietary consumption in the statistical analysis studying the association between diet and QoL. For example, the daily consumption frequency of whole grain breads, pasta, rice, porridge, muesli, etc. was summed and used as an indicator of whole grain product consumption. If the consumption frequency of a food was reported as times per week, the weekly consumption frequency was divided by seven to get the mean daily consumption frequency. Two sums of variables were also derived: the daily consumption frequency of healthy foods was the sum of the consumption frequencies in the 11 healthy food groups, while the daily consumption frequency of unhealthy foods was the sum of the consumption frequencies in the six unhealthy food groups. Regular breakfast was defined as having breakfast every day. Since we did not have the data on energy intake, we adjusted the statistical analysis for physical activity to include the effects of the energy intake in the model.

### 2.8. Statistical Analysis

The descriptive statistics were presented as means with SDs or as counts with percentages. The linearity across the four level groups of healthy foods consumed per day was evaluated using the Cochran-Armitage test (chi-square test for trend), ordered logistic regression, and analysis of variance with an appropriate contrast (orthogonal). Van der Waerden rank-based normalization [42] was used to yield standardized scores for consumption frequencies of healthy and unhealthy foods.

A possible nonlinear relationship between EUROHIS-QOL and consumption frequencies was assessed by using four-knot-restricted cubic spline regression models with knots located at the 5th, 35th, 65th, and 95th percentiles. For restricted cubic splines, also known as natural splines, knot locations were based on Harrell’s recommended percentiles [43]. Multivariate linear regression analysis was used to identify the relationship between QoL and the normalised standardised scores for consumption of (healthy and unhealthy) foods with standardised regression coefficient beta (β). The beta value is a measure of how strongly the predictor variable influences the criterion variable. The beta is measured in units of SD. Cohen’s standard for beta values above 0.10, 0.30, and 0.50 represent small, moderate, and large relationships, respectively. The Finnish general population values for the EUROHIS-QOL [44] was weighted to match the age distribution of the study population. In the case of violation of the assumptions (e.g., non-normality) for continuous variables, a bootstrap-type method was used. The normality of variables was evaluated graphically and by using the Shapiro–Wilk W test. The Stata 16.1 (StataCorp LP, College Station, TX, USA) statistical package was used for the analysis.

## 3. Results

The study cohort consisted of 631 female employees, with a mean age of 49 (SD 10) years. Figure 1 displays the mean daily consumption frequency of various healthy and unhealthy food groups among the participants. The most frequently consumed healthy food groups were vegetables, fruit, and berries, whole grain products, and vegetable oils. The most infrequently consumed healthy food groups were legumes and fish. The most frequently consumed unhealthy food groups were high-fat dairy products, red meat, and sweet snacks. The sum of the daily consumption frequencies in healthy foods varied between 2 and 30, and in unhealthy foods between 0.1 and 14.

Table 2 displays the characteristics of the participants according to the quartiles of the healthy foods consumption frequency. The participants with a higher consumption frequency of healthy foods were slightly older (*p* = 0.002) and more often cohabiting (*p* = 0.036). They reported more physical activity (*p* = 0.004), had regular breakfast more often (*p* < 0.001), and smoked more rarely (*p* = 0.008) than the subjects with a lower consumption frequency of healthy foods. They also reported less symptoms of depression (*p* = 0.013) and anxiety (*p* = 0.018). Those with a higher consumption frequency of healthy foods had a higher mean systolic blood pressure level (*p* = 0.029) and a lower mean glucose concentration (*p* = 0.041) than those with a lower consumption frequency of healthy foods.

The EUROHIS-QOL mean score among the study participants was 4.07 (SD 0.52), and the consumption frequency of healthy foods was positively associated with the EUROHIS-QOL mean score (Table 2). The association remained significant when adjusted for age, BMI, education years and disease burden, smoking and PA (*p* for linearity = 0.016). Figure 2 displays the EUROHIS-QOL mean score (continuous variable) as a function of rank-based normal scores of the consumption frequencies of healthy and unhealthy foods. A significant positive association was seen between the consumption frequency of healthy foods and the EUROHIS-QOL mean score (*p* = 0.002), but the consumption frequency of unhealthy foods was not associated with the EUROHIS-QOL mean score.

Figure 3 displays the effect of the consumption of healthy and unhealthy foods on different dimensions of QoL in a multivariate linear regression analysis with standardised regression coefficient beta (β). The consumption of healthy foods had a small positive effect on physical health, psychological wellbeing, and environmental aspects of QoL. It had no effect on the domain of social relationships. The consumption of unhealthy foods had no significant effect on any of the dimensions of QoL. The differences between the effects of healthy and unhealthy food consumption were significant between all other dimensions of QoL except psychological wellbeing.

## 4. Discussion

In this study, we demonstrated that a more frequent consumption of healthy food groups was associated with better QoL among female municipal employees. The association was small but comprehensive, involving most dimensions of QoL. The most frequently consumed healthy food groups in the present study were vegetables, fruit, and berries, whole grain products, and vegetable oils. The consumption frequency of unhealthy food groups was not associated with QoL in this study.

The grouping of the food items as healthy or unhealthy was based mainly on the Nordic Nutrition Recommendations [39]. The health benefits of the Mediterranean style diet have been shown in several studies [3], but we wanted to use the Nordic recommendations, since they have been adjusted to the local cuisine. Eating according to them has previously been shown to be associated with health benefits and lower all-cause mortality [45,46].

In the present study, the participants consuming more frequently healthy foods, also had other favourable health habits. Their physical activity was higher, and they smoked more rarely compared to the participants with lower consumption rate of healthy foods. This was expected, since risk-factors and healthy habits tend to cluster [47]. In addition, the prevalence of depressive and anxiety symptoms was lower among those who ate healthy foods more often. There is also previous information that abundant use of fruit, vegetables, fish, and high-fibre cereal products is associated with better mental health measures [17,18,19,20].

The differences in the physical health metrics compared between the groups of different consumption frequencies of healthy foods were small. There was no significant difference in BMI or blood lipids, and the number of previously diagnosed chronic diseases was on a similar level. The mean systolic blood pressure was on a slightly higher level among those who ate more healthy foods. We assume that this was most likely due to their older age. The only favourable finding in physical metrics in association with healthy foods consumption frequency was that the mean concentration of fasting glucose was on a lower level among those who consumed more healthy foods. There is convincing scientific evidence that abundant use of unhealthy foods, rich in saturated fats, added sugar and salt, as well as alcohol, is associated with higher BMI, blood lipids, blood pressure, and blood glucose levels [1,48]. Presumably, more differences in physical health metrics would have been detected in comparison with consumption frequencies of unhealthy foods.

The EUROHIS-QOL mean score among the study participants (4.07) was very close to the mean score of 4.03 among age-matched healthy female controls from the Finnish national Health 2011 survey [44]. Higher consumption frequency of healthy foods was positively associated with the EUROHIS-QOL mean score, but no association between unhealthy food consumption frequency and QoL was detected in the present study. In the previous literature, recommended, balanced, diets have been linked with good QoL and well-being in many studies [9,10,11,12,13], but also unhealthy “Western” diet has been associated with poorer QoL [11]. These associations have been detected with diets consisting of a mixture of healthy and unhealthy foods, and it cannot be determined whether the results are related to the use of healthy or unhealthy components of the diet. There is also evidence about the associations of certain food groups with QoL, especially the intake of fruit and vegetables, but also the consumption of fish and whole grain products has been connected with better QoL [16,17,18,19,20]. There are also a few works where consumption of certain unhealthy food groups has been associated with poorer QoL in children and young adults [18,21], but in most studies also in this population, the association of poorer QoL has been shown with an unhealthy dietary pattern [18]. This previous information and the results in the present study support the idea that regarding QoL, adequate use of healthy food groups appears to be more important than avoiding the unhealthy food groups. The adverse associations of unhealthy diets with QoL may have been mediated mainly through insufficient use of healthy foods.

In the present study, a multivariable linear regression analysis with standardised regression coefficient beta was conducted to compare the relationships between consumption frequencies of healthy and unhealthy food groups and different dimensions of general quality of life. More frequent consumption of healthy foods had a positive effect on every other dimension of QoL except social relationships, and the consumption of unhealthy foods had no significant effect on any of the dimensions. There are few previous studies where the association of a dietary pattern or food group consumption with QoL would have been assessed using a general QoL measure, especially among general population settings. However, in a study by Lee et al. [9] among Korean adults, high adherence to a modified Korean version [49] of the Recommended Food Score [50] was also associated with better general QoL in every dimension of the WHOQOL-BREF [29] in women. In a study among Chilean university students, the consumption of fast food and sweet snacks was associated with lower scores only in the physical health domain of the WHOQOL-BREF [21]. The dimensions of WHOQOL-BREF and the EUROHIS-QOL, used in the present work, are comparable. To our knowledge, no previous studies about the association between food consumption and QoL have been conducted using the EUROHIS-QOL.

According to our results, a frequent use (compared to a more infrequent use) of food items in the food groups considered healthy in the present study, appear to have a comprehensive, positive association with good general QoL among female public sector employees. This information adds to the previous knowledge of what aspects in nutrition are the most importantly associated with QoL. Our findings might provide some new ideas for the designing of very simple diet counselling for working-age women, since the desire for well-being and good QoL are found to motivate people to make lifestyle changes [7,8]. The advice to increase the consumption of healthy foods on a rather short list would be quite simple and reasonably achievable.

The strengths of the study were that validated instruments were used to assess the food consumption [37,38] and QoL [27], and the clinical measurements were made by trained medical staff. We also acknowledge some limitations: Since only female participants were analysed, the results cannot be generalised to males. Although the validity and reliability of the EUROHIS-QOL are demonstrated previously in several countries and in different populations [30,31], they could not have been tested particularly in our study population. With FFQ we only have information about the food consumption frequencies, not about the amounts of food consumed. However, it is shown that food consumption frequencies and amounts tend to correlate quite well [38] (Unpublished data). It is well known that healthy habits tend to cluster [47], and this trend was also seen in the current study. This may have accentuated the association of healthy eating with QoL. It is known that response rates in email surveys tend to be lower than in mail surveys [51], but it is still possible that the representativeness of the study sample is compromised. It is possible that the healthiest sub-group of the workforce is also the most willing to attend voluntary health surveys, which may result in the possibility that our results reflect the situation in the mainly healthy section of the workforce. However, the mean age ratio of the study participants was comparable to the entire personnel of the city of Pori, and the gender distribution among the initial respondents resembled the distribution among the employees working at the units that were invited to the study [22,24]. The EUROHIS-QOL mean score among the study participants (4.07) was also very close to the mean score of 4.03 among age-matched healthy female controls from the Finnish national Health 2011 survey [44]. These results suggest that the participants of this study were nevertheless a representative sample of female employees.

The data of the present study are already seven years old, but it is known that health habits do not change very quickly on a population level, and the food choices defined as healthy and unhealthy in this study have been available similarly in Finland already long before and after the study was conducted. Thus, we find that the data are still valid for showing the associations of the present work. It would, however, be interesting to conduct a new study among working-age women in Finland, to determine if the food consumption habits have changed since 2015. It would be especially interesting to study the impact of the COVID-19 pandemic on the food consumption among women. In addition, since the present study was conducted among employed females, and the study design was cross-sectional, it would be necessary to confirm the results in different study designs and populations. An intervention study among working-age women would also be needed to investigate if the simple diet counselling suggested according to the results of the present study (advice to use frequently healthy food groups), would improve the diet and QoL of the participants.

## 5. Conclusions

In the present study, we managed to connect the consumption frequencies of healthy food groups with QoL, an outcome that is found to motivate people to change their lifestyles. The more frequent consumption of healthy food groups (e.g., vegetables, fruit and berries, whole grain products, vegetable oils, nuts, seeds, and fish) was associated with better results in almost all dimensions of general QoL, but the consumption frequency of unhealthy foods was not associated with QoL in this study. According to these results, we suggest that recommending abundant use of healthy foods on a simple list could be a way to perform very simple diet counselling to patients, especially in short primary health care appointments.

## Figures and Tables

**Figure 1 nutrients-14-03630-f001:**
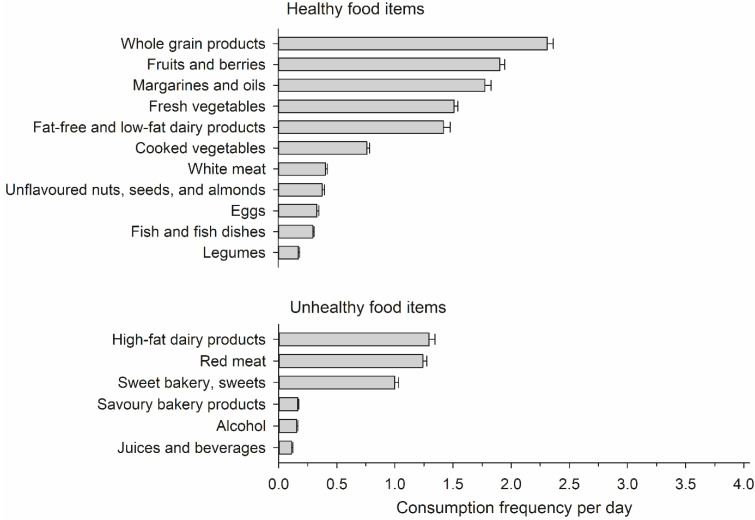
Mean daily consumption frequency for different groups of healthy and unhealthy food items. Whiskers show standard errors.

**Figure 2 nutrients-14-03630-f002:**
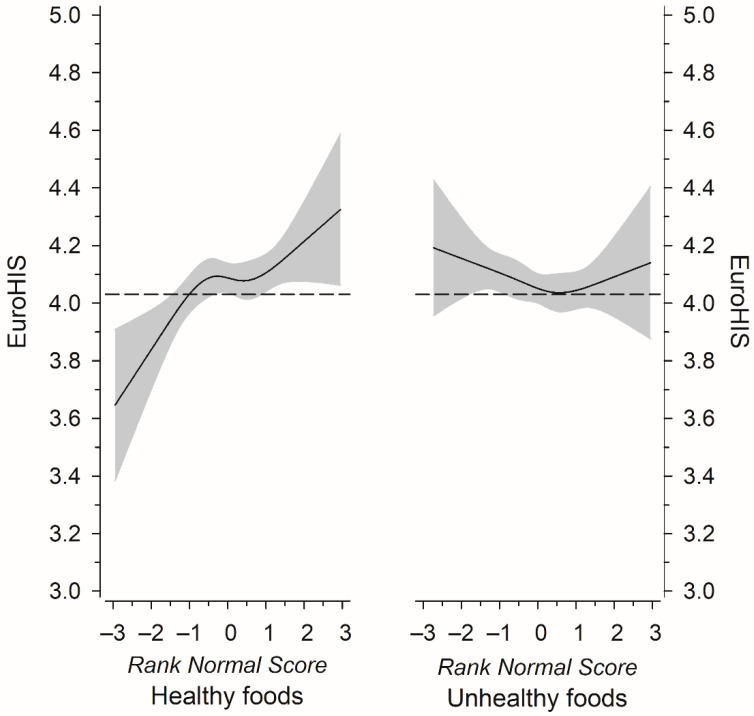
Relationships of EUROHIS-QOL 8-item Index mean as the function of rank-based normal scores of the consumption frequencies of healthy and unhealthy foods. The curves were derived from four-knot restricted cubic splines regression models. The models were adjusted for age, body mass index, education years, number of chronic diseases, smoking, and physical activity. Gray areas represent 95% confidence intervals.

**Figure 3 nutrients-14-03630-f003:**
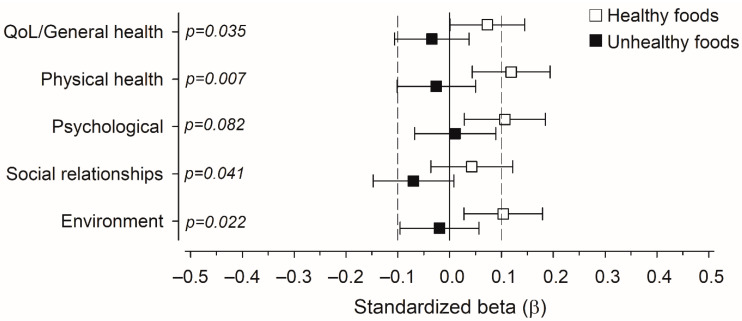
Relationships between consumption of (healthy and unhealthy) food items and different dimensions of quality of life (QoL) in a multivariate linear regression analysis with standardised regression coefficient beta (β). *p*-values are for the difference between the effect of healthy and unhealthy foods consumption. The models were adjusted for age, body mass index, education years, number of chronic diseases, smoking, and physical activity. Whiskers represent 95% confidence intervals.

**Table 1 nutrients-14-03630-t001:** Food groups considered healthy or unhealthy, based on the Nordic Nutrition Recommendations (Section 5; Food, food patterns and health outcomes—Guidelines for a healthy diet) [39] and on previous scientific understanding about the association between diet and cognitive health [40,41].

Healthy Food Groups	Unhealthy Food Groups
Fat free milk and sour milk, low-fat cheese (fat < 20%)	Red meat, sausages, red cold meat
Unflavoured nuts, seeds, and almonds	Juices and beverages sweetened with sugar
Legumes (peas, lentils, beans)	Savoury bakery products such as pies and pastries, potato chips and nachos, popcorn, salty nuts
Fresh vegetables	
Fresh fruits and berries	Sweet bakery products (buns, pies, cookies, cakes), chocolate, sweets
Whole grain pasta and rice, rye bread, rye crisp bread, breakfast cereal, muesli, porridge	Alcohol
Fish and fish dishes	High-fat dairy products: full fat milk and sour milk, full-fat cheese (fat > 20%), butter, butter-oil spreads (fat > 80%)
Margarines and oils (cooking, bread spread, salad dressing)	
Cooked vegetables	
Eggs	
White meat	

**Table 2 nutrients-14-03630-t002:** Characteristics of the participants according to the quartiles of the consumption frequencies of healthy foods.

	Quartiles of Healthy Foods Consumed Per Day				*p* for Linearity
	I (<7.5) n = 126	II (7.5–10.4) n = 189	III (10.5–14.7) n = 189	IV (>14.7) n = 126	
**Sociodemographic factors**					
Age, years, mean (SD)	48 (10)	47 (9)	49 (10)	52 (9)	0.002
Education years, mean (SD)	13.7 (2.7)	14.1 (2.6)	14.0 (2.8)	14.1 (2.7)	0.52
Cohabiting, n (%)	94 (75)	149 (79)	156 (83)	106 (84)	0.036
**Health behaviours**					
PA, hours per week, mean (SD)	2.0 (3.5)	2.5 (2.2)	2.8 (3.2)	2.9 (2.2)	0.004
Good quality of sleep, n (%)	87 (69)	157 (83)	139 (74)	96 (76)	0.68
Regular breakfast, n (%)	98 (78)	173 (92)	176 (93)	115 (91)	<0.001
AUDIT-C, mean (SD)	3.0 (1.6)	2.7 (1.6)	2.7 (1.7)	2.7 (1.5)	0.11
Current smoking, n (%)	19 (15)	15 (8)	10 (5)	8 (7)	0.008
**Clinical factors**					
Blood pressure, mmHg, mean (SD)					
Systolic	129 (16)	130 (17)	132 (18)	133 (18)	0.029
Diastolic	85 (9)	84 (11)	85 (10)	84 (11)	0.95
Body mass index, kg/m^2^, mean (SD)	26.9 (5.3)	27.2 (5.2)	26.6 (4.6)	26.2 (4.3)	0.11
Total cholesterol, mmol/L, mean (SD)	5.40 (0.99)	5.22 (0.85)	5.16 (0.94)	5.36 (0.88)	0.58
LDL cholesterol, mmol/L, mean (SD)	3.11 (0.78)	2.98 (0.71)	2.89 (0.77)	3.03 (0.69)	0.24
HDL cholesterol, mmol/L, mean (SD)	1.76 (0.42)	1.76 (0.43)	1.80 (0.45)	1.84 (0.48)	0.11
Triglycerides, mmol/L, mean (SD)	1.20 (0.57)	1.09 (0.57)	1.06 (0.56)	1.08 (0.59)	0.12
Fasting glucose, mmol/L, mean (SD)	5.60 (0.56)	5.45 (0.46)	5.51 (0.64)	5.43 (0.50)	0.041
Anxiety, GAD-7, mean (SD)	3.4 (3.3)	2.7 (3.2)	3.3 (3.7)	2.1 (2.7)	0.018
Major Depression Inventory (MDI), mean (SD)	6.5 (6.3)	4.2 (4.9)	6.1 (6.7)	3.7 (4.1)	0.013
Number of chronic diseases, mean (SD)	1.2 (1.3)	0.9 (1.0)	1.4 (1.4)	1.2 (1.2)	0.13
EUROHIS-QOL mean score, mean (SD)	3.95 (0.57)	4.13 (0.46)	4.01 (0.53)	4.19 (0.47)	0.010

AUDIT-C, The 3-item Alcohol Use Disorders Identification Test; GAD-7, General Anxiety Disorder 7-item Scale; EUROHIS-QOL, EUROHIS QOL 8-item Index; HDL, High density lipoprotein; LDL, Low density lipoprotein; PA, Physical activity; SD, Standard deviation.

## Data Availability

The data that support the findings of this study are available from the corresponding author upon reasonable request.

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
