# Peer review of "Healthy and Unhealthy Food Consumption in Relation to Quality of Life among Finnish Female Municipal Employees: A Cross-Sectional Study"

_nutrients, 2022, doi:10.3390/nu14173630_

Round 1
Reviewer 1 Report
The article concerns an important research problem and I assess it positively. I recommend the following changes to improve its quality:
1. In the Introduction, a synthetic review of the literature on the potential correlations of nutrition with the quality of life should be presented.
2. In the Methods chapter, the application of linear regression should be justified in the context of the distribution of variables and the type of measurement scales used.
3. A research gap should be identified.
4. There is no attempt to explain the obtained research results in the Discussion.
5. The limitations and contribution of research to the scientific system have not been indicated.
6. There are no cognitive and practical conclusions regarding the quality of life of the respondents in the Conclusions chapter.
Author Response
Response to the Reviewer 1
We are very grateful of Your valuable comments and suggestions to our manuscript, which has now been revised accordingly. Please find below our responses to your specific concerns:
- In the Introduction, a synthetic review of the literature on the potential correlations of nutrition with the quality of life should be presented.
The Introduction section has now been extended, and more potential correlations between nutrition and quality of life have been described.
- In the Methods chapter, the application of linear regression should be justified in the context of the distribution of variables and the type of measurement scales used.
The justification of linear regression is now clarified in the Statistical analysis section.
- A research gap should be identified.
A research gap is now identified at the end of the Introduction section, before the study aim.
- There is no attempt to explain the obtained research results in the Discussion.
The Discussion section has now been rewritten to better explain the results and to compare them with previous literature.
- The limitations and contribution of research to the scientific system have not been indicated.
We have now tried to better explain the contribution of our research to scientific system, and there are also some adjustments in the limitation part.
- There are no cognitive and practical conclusions regarding the quality of life of the respondents in the Conclusions chapter.
The Conclusion chapter has now been revised to better conclude the findings regarding the quality of life of the respondents.
Reviewer 2 Report
This manuscript reports an interesting field experiment of the healthy and unhealthy food consumption in relation to quality of life in 2014 and 2015. The data they obtained are precious. It should be published if the authors would like to make a revision.
1 My suggestion is that, if possible, more up-to-date data could be used in order to observe whether there have been any changes in healthy and unhealthy food consumption in relation to quality of life.
2 As a general recommendation, the authors should examine if the pandemic has impact on the choose of healthy and unhealthy food consumption.
3 The discussion part is not well written. The authors should have more discussion on the comparison between the data of Finland with other countries, on the basis of data collection using similar methods.
Author Response
Response to the Reviewer 2
We want to thank you for your valuable comments and suggestions about our manuscript, which has been revised accordingly. Please find below our responses to your specific concerns:
- My suggestion is that, if possible, more up-to-date data could be used in order to observe whether there have been any changes in healthy and unhealthy food consumption in relation to quality of life.
Unfortunately, we do not have more up-to-date data from this study. Since health habits do not change very quickly on a population level and the food groups have been available similarly in Finland already long before and after the study was conducted, we find the data still valid for this study. However, it would be really interesting to conduct a new study to find out if the food choices have changed in seven years, and if the pandemic has somehow affected the choices of healthy and unhealthy foods. This information has been added to the Limitations of the study.
- As a general recommendation, the authors should examine if the pandemic has impact on the choose of healthy and unhealthy food consumption.
Please, see above.
- The discussion part is not well written. The authors should have more discussion on the comparison between the data of Finland with other countries, on the basis of data collection using similar methods.
The Discussion section has now been rewritten to better explain the results and to compare them with relevant previous literature.
Reviewer 3 Report
1. The literature review is weak. The association between food consumption and QoL is related to many areas such as health, food, nutrition, sociology, etc. Thus, there is much research on this topic. Accordingly, I suggest revisiting the previous studies for providing a better understanding of the research gap, research questions, and novelty of this study. For this, it would be great to separate the introduction section and literature review.
2. The data collection process of the 2015 study should be described. Please provide details about the population, sample, sampling technique, data collection process, etc.
3. Although the reliability and validity of the scales are done in many countries as the authors stated, they should be tested and reported for this study for your sample.
4. The data from 2015 was quite old. Thus, the research should be renewed.
5. The discussion is very limited. Please discuss your findings with previous studies in order to reveal similarities and differences.
6. The contribution is not clear. What is your contribution to the existing literature? Please also provide implications for practice. Limitations and suggestions for further research need to be explained.
Author Response
Response to the Reviewer 3
We are very grateful of your valuable comments and suggestions to our manuscript, which has now been revised accordingly. Please find below our responses to your specific concerns:
- The literature review is weak. The association between food consumption and QoL is related to many areas such as health, food, nutrition, sociology, etc. Thus, there is much research on this topic. Accordingly, I suggest revisiting the previous studies for providing a better understanding of the research gap, research questions, and novelty of this study. For this, it would be great to separate the introduction section and literature review.
The introduction part has now been extended and improved to better describe the previously known associations between nutrition and quality of life. A research gap is now also better defined. The paragraphs of the introduction section are organised to better separate the introduction text and the literature review.
- The data collection process of the 2015 study should be described. Please provide details about the population, sample, sampling technique, data collection process, etc.
The data collection processes (2014 and 2015) are now described in detail.
- Although the reliability and validity of the scales are done in many countries as the authors stated, they should be tested and reported for this study for your sample.
This is a good point, but unfortunately, would have needed a separate study and could not have been performed here. This information has been added to the limitation section.
- The data from 2015 was quite old. Thus, the research should be renewed.
It is true that the data are seven years old, but since health habits do not change very quickly on a population level and the food groups have been available similarly in Finland already long before and after the study was conducted, we find the data still valid for this study. This information is, however, added to the limitations of the study.
- The discussion is very limited. Please discuss your findings with previous studies in order to reveal similarities and differences.
The Discussion section has now been rewritten and extended to better explain the results and to compare them with previous literature.
- The contribution is not clear. What is your contribution to the existing literature? Please also provide implications for practice. Limitations and suggestions for further research need to be explained.
We have now clarified the contribution to the existing literature and provided practical implications. There are some adjustments also in the limitation part, and suggestions for further research have been added.
Reviewer 4 Report
The submitted manuscript aims at examining whether consumption frequencies of healthy and unhealthy food items were associated with the quality of life (QoL) in a group of 631 female employees from the city of Pori, Finland. The Authors hypothesised that a frequent consumption of healthy food groups would have a positive relationship on QoL.
The Introduction is very short and would benefit from a wider characteristics the EUROHIS-Quality of Life 8-item index which was used to assess the QoL.
In the Material and Methods section (L72-74) it is explained that (…) A total of 836 employees (104 males, 732 females) participated 72 in the study in 2014 and the response rate was 32.5%. Complete information about data collection from that year has been described earlier [15]. This is confusing because the presented study was conducted in 2015 (L23-24) while the results of the cross-sectional study published in 15. Veromaa et al. Scand. J. Public Health 2017 had been conducted among 732 female employees in 2014. This means that the two described groups were different, besides if publication 15 is not open access I would suggest a more detailed description of the data collection, including months as seasonality also can have a big impact on consumption frequencies (this can also be mentioned among limitations (294-312)
L164 – Table 1 – due to the fact that the terms “healthy” and “unhealthy” foods are often discussed please specify in which part of the 2012 Nordic Nutrition Recommendations foods are classified as healthy and unhealthy.
L286-288 Can you specify what is – according to Authors - frequent consumption of healthy foods (to have has impact on QoL)
L314-315 and L319 I suggest reconsidering using the words “rather simple” in the Conclusions. Please give examples of “healthy” (recommended by guidelines) food products and point out which groups were consumed most frequently and which least (legumes) in the studied group of women. Please check if you mean “legumes” or “pulses” (legumes include soya – for reference see Introduction to Nutrients | Free Full-Text | Towards More Sustainable Diets—Attitudes, Opportunities and Barriers to Fostering Pulse Consumption in Polish Cities (mdpi.com)
Author Response
Response to the Reviewer 4
We want to thank you for your valuable comments and suggestions about our manuscript, which has been revised accordingly. Please find below our responses to your specific concerns:
- The Introduction is very short and would benefit from a wider characteristics the EUROHIS-Quality of Life 8-item index which was used to assess the QoL.
The introduction has now been extended, and more potential correlations between nutrition and quality of life have been described. More information about the EUROHIS-QOL 8-item index have been added to the Methods section, and in the Discussion part, a sentence “To our knowledge, no previous works about the association between food consumption and QoL has been conducted using the EUROHIS-QOL.” has been added.
- In the Material and Methods section (L72-74) it is explained that (…) A total of 836 employees (104 males, 732 females) participated 72 in the study in 2014 and the response rate was 32.5%. Complete information about data collection from that year has been described earlier[15]. This is confusing because the presented study was conducted in 2015 (L23-24) while the results of the cross-sectional study published in 15. Veromaa et al. Scand. J. Public Health 2017 had been conducted among 732 female employees in 2014. This means that the two described groups were different, besides if publication 15 is not open access I would suggest a more detailed description of the data collection, including months as seasonality also can have a big impact on consumption frequencies (this can also be mentioned among limitations (294-312)
We have now added some more detailed information about data collection both in 2014 and 2015, and provided information about comparisons between these two study populations.
The data collection was performed both at baseline and at follow-up in October – December. This part of year is especially dark and quite cold here in Finland, and although the availability of fruit and vegetables is good throughout the year, the consumption frequencies might be higher during the summer. As this is a cross-sectional study, we suggest that the seasonality has no fundamental effect on the results, since all the data are from a rather short period in late autumn / early winter.
- L164 – Table 1 – due to the fact that the terms “healthy” and “unhealthy” foods are often discussed please specify in which part of the 2012 Nordic Nutrition Recommendations foods are classified as healthy and unhealthy.
We have added the references also to the Table 1 capture. The division is based on the section 5; Food, food patterns and health outcomes – Guidelines for a healthy diet (pp. 103 - 136), especially pp.122 – 124 in the Nordic Nutrition Recommendations. https://www.norden.org/en/publication/nordic-nutrition-recommendations-2012 Some adjustments for the list (for example the inclusion of eggs in the healthy foods) was made according to the results in previous studies concerning diet and cognitive health. In the Nordic recommendations, eggs are not regarded as unhealthy, but more like neutral nutrients.
- L286-288 Can you specify what is – according to Authors - frequent consumption of healthy foods(to have has impact on QoL)
We have now added a specification: a frequent use (compared to a more infrequent use) of healthy foods (L360 in the revised version).
- L314-315 and L319 I suggest reconsidering using the words “rather simple” in the Conclusions. Please give examples of “healthy” (recommended by guidelines) food products and point out which groups were consumed most frequently and which least (legumes) in the studied group of women. Please check if you mean “legumes” or “pulses” (legumes include soya – for reference see Introduction to Nutrients | Free Full-Text | Towards More Sustainable Diets—Attitudes, Opportunities and Barriers to Fostering Pulse Consumption in Polish Cities (mdpi.com)
The conclusion has now been revised, and we have also provided some examples of the recommended healthy foods. We have also added the information about which were the most frequently and infrequently consumed food groups among the study participants. This information is provided in the results and in the first paragraph of the discussion section. Soy was also included in the group of legumes in our study, so the definition has not been changed.
Round 2
Reviewer 1 Report
The authors made the suggested corrections. I am in favor of publishing the article.
Reviewer 3 Report
Thank you for your significant improvements. Although the paper is promising in its current form, I suggest improving the theoretical and managerial implications by providing more specific suggestions and explanations.